# Poor Self-Rated Sleep Quality and Quantity Associated with Poor Oral Health-Related Quality of Life among Indigenous Australian Adults

**DOI:** 10.3390/ijerph21040453

**Published:** 2024-04-08

**Authors:** Xiangqun Ju, Joanne Hedges, Sneha Sethi, Lisa M. Jamieson

**Affiliations:** Australian Research Centre of Population Oral Health, Adelaide Dental School, University of Adelaide, Adelaide 5000, Australia; joanne.hedges@adelaide.edu.au (J.H.); sneha.sethi@adelaide.edu.au (S.S.); lisa.jamieson@adelaide.edu.au (L.M.J.)

**Keywords:** sleep quality, sleep quantity, OHRQoL, Indigenous

## Abstract

Background: Indigenous Australians score worse on both sleep and oral health. This study aimed to evaluate sleep quality and quantity associated with oral health-related quality of life (OHRQoL) among Indigenous Australian adults. Methods: A cross-sectional study involving 728 Indigenous Australian adults aged 18+ years was conducted. Exposure variables were sleep quality and quantity. The primary outcome variable was Oral Health Impact Profile-14 (OHIP14), which has been used to assess OHRQoL. Multivariable log–Poisson regression models were applied to estimate the mean ratios (MRs) for mean OHIP14 scores. Results: The average OHIP14 score was 14.9, and the average amount of sleep was 6.8 h/night. After adjusting for all covariates, self-rated very bad sleep quality was associated with 2.2 times (MR = 2.17, 95% CI: 1.97–2.37) higher OHIP14 scores than those who rated their sleep quality as very good. Participants who self-reported sleeping 7–8 h/night had 0.9 times (MR = 0.89, 95%CI: 0.83–0.95) lower OHIP14 scores than those sleeping more than 8 h. Conclusions: The average number of sleep hours for Indigenous participants were lower than recommended (7–8 h/night). Our findings indicate that poor sleep quality and quantity, and oral health-related behaviours associated with sleep deprivation were positively associated with poor oral health related quality of life among Indigenous Australian adults.

## 1. Introduction

Sleep is generally considered a pillar of good health. Sleep is a fundamental operating state of the central nervous system and is important for physical and psychological health. Sleep quality means that a person can fall asleep quickly, cycle through light and deep sleep stages and get full duration in these stages [1,2]. A sleep duration of 7–8 h/night is considered appropriate for optimal mental and physical health [3,4]. Previous studies have suggested that poor sleep quality and deprivation (<7 h/night) might influence and be affected by many chronic diseases, such as diabetes, hypertension, cardiovascular disease [5] and immune system disorders [6].

Oral health refers to the health of the entire oral-facial, teeth and gum regions that allows an individual to speak, smile, taste, touch, chew and swallow food. Oral health is considered a mirror of, and closely related to, overall health (including sleep health), well-being and quality of life [7]. Oral health-related quality of life (OHRQoL) as a part of quality of life can be measured by using a valid and reliable non-clinical standardized instrument—the Oral Health Impact Profile 14 (OHIP14) [8].

The first peoples of Australia, Aboriginal and Torres Strait Islanders (hereafter respectfully termed ‘Indigenous’), comprise 3.2% of the total Australian population [9]. Previous studies have shown that Indigenous Australians score worse not only on general health [10] but also on oral health and sleep health [11,12] than their non-Indigenous counterparts. Evidence has shown [12] that Indigenous Australians are significantly more likely to report worse sleep health than non-Indigenous Australians in all measured domains of sleep, and nearly one-third of Indigenous Australians experience less than the recommended 7–8 h/night.

Evidence shows that sleep disorders, such as obstructive sleep apnea (OSA), and poor sleep quality are caused by oral and dental diseases, such as dentognathic deformity, tooth loss and periodontal disease [6,13,14,15,16,17,18]. But there has been minimal investigation of whether poor sleep health would affect oral health and oral health-related quality of life. Despite individual efforts and small independent studies to ascertain the relationship between the two, to the best of our knowledge, there have been no population-based studies examining the relationship between oral health status and sleep quality and duration in Indigenous adults. The aim of this study was to evaluate the sleep quality and quantity associated with OHRQoL among Indigenous Australian adults. We have hypothesized that sleep quality and quantity would interfere with oral- and dental-related behaviours and affect OHRQoL

## 2. Materials and Methods

Strengthening the Reporting of Observational Studies in Epidemiology (STROBE) guidelines were used to report the study.

### 2.1. Study Design, Sample Selection and Data Collection

A cross-sectional sub-study design and data were obtained from the 24-month follow-up of a longitudinal cohort study of Indigenous Australians. A large convenience sample (*n* = 1011) of Indigenous Australian adults in South Australia aged 18+ years was recruited through Aboriginal Community Controlled Health Organisations (ACCHOs) and supervised by an Indigenous Reference Group (IRG).

#### Sample Size Calculation

When using a mean = 1.64 (SD = 2.44) or prevalence = 16.9% of ‘Occasionally, fairly often or very often’ OHIP14 [8] (exposed/non-exposed rate = 1:6), type I error = 5%, type II error = 20%, 80% test power, and power to detect a risk ratio = 2 with a 95% confidence level, the necessary sample was estimated to be 99 to 246 individuals. An additional 30% was added to compensate for losses and missing value, giving a final calculated number of 129 to 320 individuals.

### 2.2. Study Variables

#### 2.2.1. Exposure Variables

The exposure variables were sleeping status and lack of sleep-related oral health behaviours:

#### Sleep Status

Sleep quality was obtained from the question “During the past month, how would you rate your sleep quality overall?”, and the response in order of best to worst to ‘1 = Very good’, ‘2 = Fairly good’, ‘3 = Fairly bad’ or ‘4 = Very bad’.Sleep quantity (duration) was derived from the question “During the past month, how many hours of actual sleep did you get at night? This may be different to the number of hours you are in bed”. These were categorised as ‘<7 h’, ‘7–8 h’ or ‘>8 h’.Degree of snoring: This variable stemmed from the question ‘Do you snore loudly? (Loud enough to be heard through closed doors OR your bed partner elbows you for snoring at night OR you may sleep in a room with family and been told that you snore)’ and the response was dichotomized as ‘Yes’ or ‘No’.

#### Lack of Sleep-Related Oral Health Behaviours

These variables were derived from the questions “Does lack of sleep: (1) reduce your ability to brush your teeth (morning and night); (2) increase your desire to eat sugary foods or drink?” and “Do you consume sugary drinks to help you stay awake?”, and the responses were classified into ‘Often/always’, ‘Sometimes’ or ‘Never/rarely’.

#### 2.2.2. Outcome Variable

The outcome variable was the Oral Health Impact Profile-14 (OHIP14), which was used to assess OHRQoL. OHIP14 was derived from the question “During the last year, have you had/felt/found/…because of problems with your teeth, mouth, or dentures?” Each OHIP14 item was scored using a 5-point Likert scale ranging from 1 to 5 (1 = Very often…5 = Never). After re-coding of 5 to 0, 4 to 1, 3 to 2, 2 to 3 and 1 to 4, summary scores ranged from 0 to 56. High scores indicate a high oral health impact and lower oral health-related quality of life.

#### 2.2.3. Covariates

##### Sleep Health-Related Chronic Diseases

These two chronic diseases stemmed from the question “Do you have or are you being treated for (1) high blood pressure and (2) diabetes?” and was dichotomized as ‘Yes’ or ‘No’.

##### Health-Related Behaviours

Health-related behaviours included ‘Tobacco and non-tobacco smoking statuses’ and were defined as ‘current smoker’, ‘ex-smoker’ or ‘never smoked’. ‘Consume alcohol’ was categorized as ‘Daily, Weekly, Monthly or Never’. ‘Recreational drug use’ was classified as ‘Currently use’, ‘Don’t now but used to’ or ‘Never used’.

##### Social Demographic Characteristics

‘Age’ was grouped into ’18–34′, ’35–54′ or ’55 or over’ years; ‘Gender’ was ‘Male’ or ‘Female’; ‘Residential location’ was defined as ‘Non-metropolitan’ (regional areas) or ‘Metropolitan’; ‘Highest educational level’ was categorized as ‘High school or less’ vs. ‘Trade/TAFE/University’, where ‘Technical and Further Education (TAFE)’ represents training for vocational occupations; ‘Income’ was defined by total annual household income, which was dichotomized into ‘Job’ vs. ‘Centrelink, which provides welfare payments to those who are unemployed’; Health care card (HCC) is means-tested and enables access to services such as publicly funded dental care and classified ‘Yes’ or ‘No’; Car ownership was derived from the question ‘Do you own a car?’ with two possible responses, ‘Yes’ or ‘No’.

### 2.3. Statistical Analyses

#### 2.3.1. Descriptive Analyses

The analysis began with the computation of univariate statistics describing the frequency and percentage of covariates and associated with the severity (mean and 95% confidence interval (CI)) of OHIP14. Statistically significant differences were denoted by 95% CIs that did not overlap.

The reliability of self-rated sleep quality was measured by using Cronbach’s α and corrected item total correlations (CITCs) to assess internal consistency. Internal consistency was acceptable when Cronbach’s standardized alpha (α) was ≥0.70 [19,20]. A CITC value lower than 0.30 indicates that an item displays poor internal consistency with the other items and should be excluded.

#### 2.3.2. Bivariate Analysis

Bivariate analysis and Student’s *t*-test were used to analyse the association between OHIP14, sleep quality and sleep hours. A *p*-value of 0.05 was taken as the significance level.

#### 2.3.3. Multivariable Analyses

Multivariable log–Poisson regression models with robust standard error estimation were applied as a statistical model to estimate the multivariable relationships of OHIP14 and other covariates. Mean ratios (MRs) with their 95%CI were calculated for mean OHIP14. The dependent variable of these models was OHIP14 severity. Unadjusted models (Model 1): bivariate relationships of OHIP14 and sleep health, and sleep-related oral health behaviours and other covariates. Sleep status or sleep-related oral health behaviours were entered in Model 2; general health (two chronic diseases) and health-related behaviours were added in Model 3, with the final model (Model 4) comprising all factors, including all sociodemographic factors.

Data analysis was performed using SAS statistical software (SAS 9.4, SAS Institute Inc., Cary, NC, USA).

## 3. Results

The analyses presented involve the 728 Indigenous Australian adults who completed a questionnaire at the 24-month follow-up.

Table 1 shows the sample characteristics and associations with OHIP14 among Indigenous Australian adults. A higher proportion of participants were in the middle age group (35–54 year, 39%), female (69%), resided in non-metropolitan locations (63%), reported a highest level of education as high school or less (65%), received their income from Centrelink (75%), had an HCC (79%), owned a car (57%), currently smoked tobacco (nearly 60%), never used tobacco (68%), never consumed alcohol (more than 35%), and never used recreational drugs (approximately 50%). About 30% and 25% had hypertension and diabetes, respectively. The average number of sleep hours was 6.8 (95% CI: 6.7–7.0) per night. A higher proportion of participants self-reported ‘Fairly good’ sleep quality (nearly 50%), slept less than 7 h (more than 45%) and slept without snoring (more than 50%). The majority of participants self-reported ‘never or rarely’ to whether lack of sleep reduced the ability to brush teeth (more than 65%), whether lack of sleep increased the desire to eat sugary food or drinks (nearly 50%) and whether they consumed sugary drinks to help stay awake (approximately 70%), respectively.

The average OHIP14 score for the participants was 14.9 (95% CI: 13.9–15.8). Higher OHIP14 scores were observed among current smokers (16.7) and current non-tobacco smokers (18.9), those with hypertension (17.1), those who self-rated very bad sleep quality (20.8), those sleeping less than 7 h on average per night (16.9) and those reporting ‘often or always’ to whether lack of sleep reduced their desire to brush teeth (21.2), increased the desire to eat sugary food or drinks (20.1) or they consumed sugary drinks to help stay awake (22.0) (Table 1).

Table 2 shows the results of reliability analysis for self-rated sleep quality. Cronbach’s standardized alpha was more than 0.9, which appeared to have good internal consistency; the total item correlation values were more than 0.9, indicating that each item correlates well with the scale overall.

Table 3 presents the association between sleep quality and quantity, and OHIP14. OHIP14 scores increased by 4.28 with sleep quality every backward step from ‘Very good’ to ‘Very bad’. OHIP14 scores decreased by 1.09 with every increase in sleep hours on average per night. In other words, poor sleep quality and quantity was associated with poor oral health-related quality of life.

The multivariable analyses between sleep status and lack of sleep-related oral health associated with mean OHIP14 among Indigenous Australian adults are shown in Table 4 and Table 5. Higher mean OHIP14 scores were observed among those who had very bad sleep quality (MR = 2.39, 95% CI: 2.21–2.50), who slept less than 7 h per night on average (MR = 1.30, 95% CI: 1.23–1.37) and who snored (MR = 1.10, 95% CI:1.06–1.14) (Table 4). Participants who self-reported ‘often or always’ had 1.7 times higher mean OHIP14 scores than those who self-reported ‘never or rarely’ to whether lack of sleep reduced the desire to brush teeth, increased the desire to eat sugary food or drinks and they consumed a sugary drink to help stay awake, respectively (Table 5). Higher mean OHIP14 scores were observed among those who had hypertension and diabetes, were middle-aged (35–54 years) and HCC holders, who received their income through Centrelink, did not own a car, were current smokers, current non-tobacco smokers and current recreational drug users, and consumed alcohol daily. Lower mean OHIP14 scores were observed among males and those residing in non-metropolitan locations.

After adjusting for all covariates, the participants who self-rated very bad sleep quality had 2.2 times (MR = 2.17. 95% CI: 1.97–2.37) higher OHIP14 scores than those who self-rated very good sleep quality; those who self-reported sleeping 7–8 h per night on average had 0.9 times (MR = 0.89, 95% CI: 0.83–0.95) lower OHIP14 scores than those who slept more than 8 h (Table 3). OHIP14 scores were higher among those who self-reported ‘often or always’ (MR = 1.20, 95% CI: 1.12–1.29) and ‘sometimes’ (MR = 1.22, 95% CI: 1.16–1.28) than those self-reported ‘never or rarely’ to whether lack of sleep reduced the desire to brush teeth; who self-reported ‘often or always’ (MR = 1.43, 95% CI: 1.34–1.53) and ‘sometimes’ (MR = 1.34, 95% CI: 1.27–1.41) than those who self-reported ‘never or rarely’ to whether lack of sleep increased the desire to eat sugary food or drinks; and who self-reported ‘often or always’ (MR = 1.27, 95% CI: 1.19–1.36) and ‘sometimes’ (MR = 1.09, 95% CI: 1.03–1.16) than those who self-reported ‘never or rarely’ to whether they consumed a sugary drink to help stay awake (Table 4). Other risk factors related to OHIP14 scores were being middle-aged, female, residing in non-metropolitan locations, lower education level, current smoker, current and ex-non-tobacco smoker, current and ex- recreational drug user, having hypertension and diabetes, and non-ownership of a car (Table 4 and Table 5).

## 4. Discussion

The average amount of sleep per night for Indigenous Australian adults was less than the recommended 7–8 h/night (it was 6.8 h/night). Our study found a strong association between sleep health (sleep quality and quantity) and oral health-related quality of life, with poor sleep health being positively associated with poor OHRQoL (higher OHIP14 scores). The findings held even after adjusting for sociodemographic, general and oral health-related behaviours. In addition, poor oral health-related behaviours caused by lack of sleep were associated with higher OHIP14 scores.

Our findings indicated that short sleep duration, sleep deprivation and poor oral health-related behaviours caused by lack of sleep have direct or indirect negative impacts on OHRQoL. Participants with short sleep durations could be too tired to brush their teeth or might need to consume more sugary food or drink for energy. If this continues in the long term, it will affect oral health directly. Evidence has shown that poor oral hygiene and/or eating more sugary foods and drinks increases the risk of dental caries [21] and periodontitis [22] and decreases oral health-related quality of life [23]. Twice-daily tooth brushing with a fluoride-containing dentifrice is the most effective method for removal of interdental microbial plaque to prevent the onset of gingivitis and inter-dental caries; therefore, it is important for the preservation of oral health [24]. The freer the sugar consumed in food and drink, the greater the risk of tooth decay and gum disease. Sugar interacts with bacterial plaque to produce acid, which slowly dissolves the enamel, creating holes or cavities in the teeth [25]. If sleep deprivation continues, the immune system will eventually become vulnerable to infection, and indirectly, oral health behaviours will be hindered, leading to oral diseases [26]. Sleep and immunity are bidirectionally linked. The vitality of the immune system affects sleep, and sleep in turn influences the innate and adaptive arm of our body’s defence system. There is no doubt that an infection makes us tired and increases our desire to sleep, and a good night’s sleep is the best medicine for an infectious disease [27].

### Strengths and Limitations

The strengths of the study were (1) a large sample size (*n* = 724) with engagement with the Indigenous Australian community through partnerships and involvement of the study’s Indigenous Reference Group, which represented the broader South Australian Indigenous population. This means the findings potentially have meaning for other Indigenous groups residing in similar socio-economic regions of the world; (2) omnidirectional estimate sleep health, including not only used sleep quality but also sleep quantity (duration), associated with OHRQoL; and (3) adjusted large risk factors associated with both sleep and oral health. Limitations of this study were (1) the cross-sectional design, preventing the testing of causal hypotheses, and (2) non-availability of oral clinic examination data.

To the best of our knowledge, this is the first study to assess sleep quality and quantity, and to estimate lack of sleep-related oral health behaviours associated with oral health-related quality of life among Indigenous Australian adults. Our findings provide the best evidence for lack of healthy sleep being related to unhealthy sleep-related oral health behaviours, eventually leading to poor OHRQo, and make important contributions to the field of sleep health literacy. From our study findings, it is recommended that regular sleep (7–8 h/night) with oral hygiene instruction be a first-line modality to improve both sleep and oral health. Meanwhile, it is important to avoid some behaviours that are harmful to oral health, such as reducing brushing frequency and increasing the consumption of sugary food and drinks to help stay awake due to lack of sleep.

## 5. Conclusions

The average sleep hours for Indigenous Australian adults were lower (6.8 h/night) than recommended (7–8 h/night). Our findings indicate that poor sleep quality and quantity and oral health-related behaviours associated with sleep deprivation were associated with poor oral health-related quality of life among Indigenous Australian adults.

## Figures and Tables

**Table 1 ijerph-21-00453-t001:** Sociodemographic characteristics, sleep health and related oral health behaviours, and Oral Health Impact Profile-14 (OHIP14) among Indigenous Australian adults.

	N	% (95% CI)	Mean (95% CI) of OHIP14
Total	728	100	14.9 (13.9–15.8)
**Sociodemographic characteristics**			
Age group (years)			
18–34	186	25.5 (22.4–28.7)	14.5 (12.5–16.4)
35–54	286	39.3 (35.7–42.8)	15.8 (14.2–17.4)
≥55	256	35.2 (31.7–38.6)	14.1 (12.5–15.7)
Gender			
Male	226	31.0 (27.7–34.4)	13.6 (12.0–15.2)
Female	502	69.0 (65.6–72.3)	15.4 (14.2–16.7)
Residential location			
Non-metropolitan	457	62.9 (59.4–66.5)	14.3 (13.0–15.5)
Metropolitan	269	37.1 (33.5–40.6)	15.8 (14.3–17.4)
Education level			
High school or less	466	65.1 (61.6–68.6)	14.8 (13.5–16.1)
Trade or over	250	34.9 (31.4–38.4)	14.8 (13.3–16.3)
Income			
Centrelink	536	74.9 (71.7–78.0)	15.2 (14.1–16.4)
Job	180	25.1 (22.0–28.3)	13.6 (11.7–15.4)
Health care card			
Yes	550	79.0 (76.0–82.1)	15.1 (13.9–16.2)
No	146	21.0 (17.4–24.0)	14.1 (12.1–16.2)
Car ownership			
No	308	42.7 (39.1–46.3)	15.7 (14.1–17.3)
Yes	413	57.3 (53.7–60.9)	14.3 (13.0–15.5)
**Health-related behaviours**			
Smoke status			
Current smoker	404	58.4 (54.7–62.1)	16.7 (15.3–18.1)
Ex-smoker	91	13.2 (10.6–15.7)	12.2 (9.6–14.8)
Never smoker	197	28.5 (25.1–31.8)	12.0 (10.3–13.6)
Non-tobacco smoking			
Current smoker	85	12.4 (9.9–14.8)	18.9 (15.6–22.2)
Ex-smoker	133	19.4 (16.4–22.3)	15.5 (13.1–17.8)
Never smoker	469	68.3 (64.8–71.8)	14.1 (12.9–15.2)
Consume alcohol			
Daily	23	3.3 (1.9–4.6)	18.7 (12.5–24.9)
Weekly	164	23.3 (20.1–26.4)	14.0 (12.0–16.0)
Monthly	251	35.6 (32.1–39.1)	14.9 (13.3–16.6)
Never	267	37.9 (34.3–41.5)	
Recreational drug use			
Current user	142	19.8 (16.9–22.7)	16.4 (14.2–18.7)
Ex-user	233	32.5 (29.1–35.9)	15.0 (13.3–16.8)
Never use	342	47.7 (44.0–51.4)	14.2 (12.8–15.6)
**Chronic diseases**			
Had hypertension			
Yes	215	29.7 (26.3–33.0)	17.1 (15.1–19.0)
No	510	70.3 (67.0–73.7)	13.9 (12.8–15.0)
Had Diabetes			
Yes	183	25.3 (22.1–28.5)	16.3 (14.3–18.4)
No	540	74.7 (71.5–77.9)	14.4 (13.3–15.5)
**Sleep status**			
Self-rated sleep quality			
Very bad	69	9.5 (7.4–11.7)	20.8 (17.2–24.5)
Fairly bad	173	23.9 (20.8–27.0)	18.6 (16.6–20.8)
Fairly good	349	48.1 (44.5–51.8)	14.1 (12.8–15.4)
Very good	134	18.5 (15.7–21.3)	8.7 (6.8–10.7)
Sleep hours on average night			
<7 h	324	45.5 (41.8–49.2)	16.9 (15.4–18.5)
7–8 h	264	37.1 (33.5–40.6)	13.0 (11.4–14.5)
>8 h	124	17.4 (14.6–20.2)	13.1 (10.6–15.5)
Snore			
Yes	360	49.7 (46.0–53.3)	15.6 (14.2–17.0)
No	365	50.3 (46.7–54.0)	14.2 (12.8–15.6)
**Lack of sleep related behaviours**			
Lack of sleep reduce brush teeth			
Often/always	77	10.6 (8.4–12.9)	21.2 (18.0–24.4)
Sometimes	168	23.2 (20.1–26.3)	18.1 (16.0–20.3)
Never/rarely	479	66.2 (62.7–69.6)	12.7 (11.6–13.8)
Lack of sleep increase to eat sugary food			
Often/always	137	18.9 (16.1–21.8)	20.1 (17.8–22.4)
Sometimes	232	32.1 (28.7–35.5)	16.4 (14.5–18.2)
Never/rarely	354	49.0 (45.3–52.6)	11.8 (10.5–13.1)
Consume sugary drink to help awake			
Often/always	84	18.9 (9.3–14.0)	22.0 (19.1–24.9)
Sometimes	136	32.1 (16.0–21.7)	17.3 (14.9–19.7)
Never/rarely	502	69.5 (66.2–72.9)	13.0 (11.9–14.2)
	Mean (95% CI)	Min	Max
Sleep hours on average night	6.8 (6.7–7.0)	2	24

**Table 2 ijerph-21-00453-t002:** Self-rated sleep quality reliability analysis.

Self-Rated Sleep Quality	^a^ CITC
Very good	0.91
Fairly good	0.98
Fairly bad	0.92
Very bad	0.95
Cronbach’s standardized alpha (α): 0.93	

Notes: ^a^ CITC: Corrected item-total correlation.

**Table 3 ijerph-21-00453-t003:** The estimate (β) and standard error (SE) of mean OHIP14 score associated with sleep quality and quantity among Indigenous Australian adults.

Sleep Status	OHIP14		
	β	SE	*p*-Value
Self-rated sleep quality	4.28	0.56	<0.0001
Sleep hours on an average night	−1.09	0.24	<0.0001

**Table 4 ijerph-21-00453-t004:** Multivariable model between sleep status and mean OHIP14 among Indigenous Australian adults (MR, 95% CI).

	Model 1	Model 2	Model 3	Model 4
**Sleep status**				
Self-rated sleep quality				
Very bad	2.39 (2.21–2.58)	2.30 (2.12–2.50)	2.34 (2.14–2.56)	2.17 (1.97–2.37)
Fairly bad	2.14 (2.00–2.29)	2.06 (1.92–2.22)	2.06 (1.91–2.22)	1.98 (1.83–2.14)
Fairly good	1.61 (1.51–1.72)	1.62 (1.52–1.73)	1.61 (1.50–1.72)	1.60 (1.49–1.72)
Very good	ref	ref	ref	ref
Sleep hours on average night				
<7 h	1.30 (1.23–1.37)	1.03 (0.97–1.10)	0.98 (0.92–1.05)	0.99 (0.93–1.06)
7–8 h	0.99 (0.94–1.05)	0.94 (0.89–1.00)	0.88 (0.83–0.94)	0.89 (0.83–0.95)
>8 h	ref	ref	ref	ref
Snore				
Yes	1.10 (1.06–1.14)	1.03 (0.99–1.07)	0.98 (0.94–1.02)	0.98 (0.94–1.02)
No	ref	ref	ref	ref
**General health**				
Had hypertension				
Yes	1.23 (1.18–1.28)		1.21 (1.15–1.27)	1.23 (1.16–1.30)
No	ref		ref	ref
Had Diabetes				
Yes	1.14 (1.09–1.19)		1.08 (1.03–1.14)	1.12 (1.06–1.18)
No	ref		ref	ref
**Health-related behaviours**				
Smoke status				
Current smoker	1.40 (1.33–1.47)		1.31 (1.25–1.38)	1.30 (1.23–1.37)
Ex-smoker	1.02 (0.95–1.10)		0.89 (0.83–0.97)	0.85 (0.78–0.92)
Never smoker	ref		ref	ref
Non-tobacco smoking				
Current smoker	1.34 (1.27–1.42)		1.26 (1.18–1.33)	1.18 (1.10–1.25)
Ex-smoker	1.10 (1.05–1.16)		1.11 (1.06–1.17)	1.11 (1.05–1.18)
Never smoker	ref		ref	ref
Consume alcohol				
Daily	1.24 (1.13–1.37)		0.95 (0.84–1.07)	0.94 (0.84–1.07)
Weekly	0.93 (0.88–0.98)		0.92 (0.87–0.97)	0.91 (0.86–0.97)
Monthly	0.99 (0.95–1.04)		0.98 (0.93–1.04)	0.98 (0.93–1.03)
Never	ref		ref	ref
Recreational drug use				
Current user	1.16 (1.10–1.22)		1.03 (0.98–1.10)	1.08 (1.02–1.15)
Ex-user	1.06 (1.01–1.11)		0.93 (0.89–0.98)	0.94 (0.89–0.99)
Never use	ref		ref	ref
**Sociodemographic characteristics**				
Age group (years)				
≥55	1.03 (0.98–1.08)			0.95 (0.89–1.01)
35–54	1.12 (1.07–1.17)			1.06 (1.00–1.12)
18–34	ref			ref
Gender				
Male	0.88 (0.85–0.92)			0.92 (0.88–0.97)
Female	ref			ref
Residential location				
Non-metropolitan	0.90 (0.87–0.94)			0.95 (0.91–0.99)
Metropolitan	ref			ref
Education level				
High school or less	1.00 (0.96–1.04)			1.08 (1.03–1.13)
Trade or over	ref			ref
Income				
Centrelink	1.12 (1.07–1.18)			1.03 (0.96–1.11)
Job	ref			ref
Health care card				
Yes	1.06 (1.01–1.12)			0.95 (0.89–1.02)
No	ref			ref
Car ownership				
No	1.10 (1.06–1.14)			1.08 (1.03–1.13)
Yes	ref			ref

Notes: Model 1: crude model; Model 2: Sleep health; Model 3: Model 2 plus adjusting for general health and health-related behaviours; Model 4: full model: Model 3 plus adjusting for the sociodemographic factors.

**Table 5 ijerph-21-00453-t005:** Multivariable model between sleep related behaviours and mean OHIP14 among Indigenous Australian adults (MR, 95% CI).

	Model 1	Model 2	Model 3	Model 4
**Lack of sleep-related behaviours**				
Lack of sleep reduces desire to brush teeth				
Often/always	1.67 (1.58–1.76)	1.31 (1.05–1.62)	1.21 (1.14–1.30)	1.20 (1.12–1.29)
Sometimes	1.43 (1.37–1.49)	1.03 (0.84–1.26)	1.19 (1.14–1.26)	1.22 (1.16–1.28)
Never/rarely	ref	ref	ref	ref
Lack of sleep increases desire to eat sugary food				
Often/always	1.70 (1.62–1.79)	1.43 (1.14–1.81)	1.45 (1.36–1.54)	1.43 (1.34–1.53)
Sometimes	1.38 (1.32–1.45)	1.09 (0.88–1.35)	1.31 (1.25–1.38)	1.34 (1.27–1.41)
Never/rarely	ref	ref	ref	ref
Consume sugary drink to help stay awake				
Often/always	1.69 (1.60–1.78)	1.25 (1.00–1.56)	1.26 (1.18–1.34)	1.27 (1.19–1.36)
Sometimes	1.33 (1.26–1.39)	1.04 (0.84–1.29)	1.10 (1.04–1.16)	1.09 (1.03–1.16)
Never/rarely	ref	ref	ref	ref
**General health**				
Had hypertension				
Yes	1.23 (1.18–1.28)		1.21 (1.15–1.27)	1.19 (1.12–1.25)
No	ref		ref	ref
Had Diabetes				
Yes	1.14 (1.09–1.19)		1.08 (1.02–1.13)	1.12 (1.06–1.18)
No	ref		ref	ref
**Health-related behaviours**				
Smoke status				
Current smoker	1.40 (1.33–1.47)		1.27 (1.21–1.34)	1.25 (1.18–1.32)
Ex-smoker	1.02 (0.95–1.10)		0.95 (0.88–1.03)	0.89 (0.83–0.97)
Never smoker	ref		ref	ref
Non-tobacco smoking				
Current smoker	1.34 (1.27–1.42)		1.31 (1.23–1.39)	1.24 (1.16–1.32)
Ex-smoker	1.10 (1.05–1.16)		1.07 (1.01–1.13)	1.06 (1.00–1.12)
Never smoker	ref		ref	ref
Consume alcohol				
Daily	1.24 (1.13–1.37)		1.18 (1.06–1.32)	1.22 (1.09–1.36)
Weekly	0.93 (0.88–0.98)		0.94 (0.89–1.00)	0.94 (0.88–1.00)
Monthly	0.99 (0.95–1.04)		1.00 (0.96–1.05)	1.00 (0.95–1.06)
Never	ref		ref	ref
Recreational drug use				
Current user	1.16 (1.10–1.22)		1.05 (1.00–1.12)	1.08 (1.02–1.15)
Ex-user	1.06 (1.01–1.11)		0.95 (0.91–1.00)	0.99 (0.94–1.04)
Never use	ref		ref	ref
**Sociodemographic characteristics**				
Age group (years)				
≥55	1.03 (0.98–1.08)			1.06 (0.99–1.14)
35–54	1.12 (1.07–1.17)			1.11 (1.05–1.17)
18–34	ref			ref
Gender				
Male	0.88 (0.85–0.92)			0.88 (0.84–0.93)
Female	ref			ref
Residential location				
Non-metropolitan	0.90 (0.87–0.94)			0.92 (0.88–0.96)
Metropolitan	ref			ref
Education level				
High school or less	1.00 (0.96–1.04)			1.06 (1.01–1.11)
Trade or over	ref			ref
Income				
Centrelink	1.12 (1.07–1.18)			1.02 (0.95–1.10)
Job	ref			ref
Health care card				
Yes	1.06 (1.01–1.12)			0.96 (0.90–1.03)
No	ref			ref
Car ownership				
No	1.10 (1.06–1.14)			1.08 (1.03–1.13)
Yes	ref			ref

Notes: Model 1: crude model; Model 2: Lack of sleep-related oral health; Model 3: Model 2 plus adjusting for general health and health-related behaviours; Model 4: full model: Model 3 plus adjusting for the sociodemographic factors.

## Data Availability

The datasets generated and/or analysed during the current study are not publicly available due to privacy issues of the participants. Data are available from the corresponding author on reasonable request.

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
