# Peer review of "Poor Self-Rated Sleep Quality and Quantity Associated with Poor Oral Health-Related Quality of Life among Indigenous Australian Adults"

_ijerph, 2024, doi:10.3390/ijerph21040453_

Round 1

Reviewer 1 Report

Comments and Suggestions for Authors

In which language you delivered the questionnaire? Did you use a cross-cultural-validated form?

Like this 

Segù M, Collesano V, Lobbia S, Rezzani C. Cross-cultural validation of a short form of the Oral Health Impact Profile for temporomandibular disorders. Community Dent Oral Epidemiol. 2005 Apr;33(2):125-30. doi: 10.1111/j.1600-0528.2005.00215.x. PMID: 15725175.

Did you obtain the approval by the Ethic Committee?

It's not clear the use of OHIP14 in relation to sleep.

Author Response

  1. In which language you delivered the questionnaire? Did you use a cross-cultural-validated form?

Like this:  Segù M, Collesano V, Lobbia S, Rezzani C. Cross-cultural validation of a short form of the Oral Health Impact Profile for temporomandibular disorders. Community Dent Oral Epidemiol. 2005 Apr;33(2):125-30. doi: 10.1111/j.1600-0528.2005.00215.x. PMID: 15725175.

  • English was used to deliver the questionnaire.
  • Data collection included face-to-face interviews and was managed by experienced senior Indigenous research officers and had the oversight of ACCHOs and an Indigenous Reference Group (IRG).
  • No, we did not use a cross-cultural validated form, but the instrument has been used many times with Indigenous Australian groups.

  1. Did you obtain the approval by the Ethic Committee?

Yes, see ‘Institutional Review Board Statement’ (Lines 289-291, page 11)

  1. It's not clear the use of OHIP14 in relation to sleep.

We have clarified and re-written why we estimated sleep quality and quantity would be associated with oral health-related quality of life (Line 54-64, page 2)

Reviewer 2 Report

Comments and Suggestions for Authors

The aims of the study are worth investigating. However considering the large number of studies within the field of sleep health, the current study does not use appropriate objective methodology to provide any novel findings. Self reported sleep profiles are useful as a first call for investigation however taking into account commonality of sleep problems, group size ought to be much larger- power calculation should be provided. I would suggest to write this study as a short paper or communication letter.

Author Response

The aims of the study are worth investigating. However considering the large number of studies within the field of sleep health, the current study does not use appropriate objective methodology to provide any novel findings. Self reported sleep profiles are useful as a first call for investigation however taking into account commonality of sleep problems, group size ought to be much larger- power calculation should be provided. I would suggest to write this study as a short paper or communication letter.

About power:

As Aboriginal and Torres Strait Islanders comprise 3.2% of the total Australian population. Census data indicates approximately 22,000 Aboriginal adults reside in South Australia. So, our large convenience sample (n=1011) represented Indigenous Australian adults.

Reviewer 3 Report

Comments and Suggestions for Authors

In this manuscript (ijerph-2802609), the authors evaluated sleep quality and quantity associated with oral health-related quality of life among Indigenous Australian adults. They concluded that poor sleep quality and quantity were associated with higher OHIP14 scores among them. In addition, poor oral health-related behaviors caused by lack of sleep were associated with higher OHIP14 scores. However, I have a particular concern about this MS:

Major Point

The authors should explain why they wanted to evaluate sleep quality and quantity associated with oral health-related quality of life among Indigenous Australian adults in a little more detail. I understood the background that Indigenous Australians have both poorer sleep and oral health. However, many readers may be unfamiliar with the reason for their deteriorated health conditions.

Minor Point

In line 166 on page 4, the author described as follows, “About 30% and 35% had hypertension and diabetes, respectively.” However, in Table 1, the percentage of “Yes” in “Had Diabetes” was 25.3%. Please check the values in the corresponding sections.

Author Response

In this manuscript (ijerph-2802609), the authors evaluated sleep quality and quantity associated with oral health-related quality of life among Indigenous Australian adults. They concluded that poor sleep quality and quantity were associated with higher OHIP14 scores among them. In addition, poor oral health-related behaviors caused by lack of sleep were associated with higher OHIP14 scores. However, I have a particular concern about this MS:

Major Point

The authors should explain why they wanted to evaluate sleep quality and quantity associated with oral health-related quality of life among Indigenous Australian adults in a little more detail. I understood the background that Indigenous Australians have both poorer sleep and oral health. However, many readers may be unfamiliar with the reason for their deteriorated health conditions.

Thanks. Let us explain why ‘sleep’ was used as independent variable:

  1. Most previous studies use sleep as dependent variable to estimate the relationship between oral health status and sleep quality.
  2. There have been ew individual efforts and small independent studies to ascertain the relationship between sleep quality and sleep quantity and oral health and health-related quality of life.
  3. During data collection, our experienced senior Indigenous research officers noticed/found that some participants did not want to brush their teeth because of being too tired to sleep or having eaten too much sugary foods or drink that kept them awake. So, we have designed some questions about ‘Lack of sleep related oral health behaviours’ and used sleep as independent variable to estimate the relationship between the two.

We have added information and hypotheses to clarify these points (Lines 54-64, page 2).

Minor Point

In line 166 on page 4, the author described as follows, “About 30% and 35% had hypertension and diabetes, respectively.” However, in Table 1, the percentage of “Yes” in “Had Diabetes” was 25.3%. Please check the values in the corresponding sections.

Sorry, it is a typo. We have corrected it.

Round 2

Reviewer 3 Report

Comments and Suggestions for Authors

In this manuscript (ijerph-2802609), the authors revised the first version, and the revised version may be acceptable to readers. However, the following minor point that I asked last time seems to have been forgotten:

Minor Point

In line 175 on page 4, the author described as follows, “About 30% and 35% had hypertension and diabetes, respectively.” However, in Table 1, the percentage of “Yes” in “Had Diabetes” was 25.3%. Please recheck the values in the corresponding sections.

Author Response

In this manuscript (ijerph-2802609), the authors revised the first version, and the revised version may be acceptable to readers. However, the following minor point that I asked last time seems to have been forgotten:

Minor Point

In line 175 on page 4, the author described as follows, “About 30% and 35% had hypertension and diabetes, respectively.” However, in Table 1, the percentage of “Yes” in “Had Diabetes” was 25.3%. Please recheck the values in the corresponding sections.

Sorry, it is a typo. We have corrected it (Line 175, page 4).